# Electron Kinetic Entropy across Quasi-Perpendicular Shocks

**DOI:** 10.3390/e24060745

**Published:** 2022-05-24

**Authors:** Martin Lindberg, Andris Vaivads, Savvas Raptis, Per-Arne Lindqvist, Barbara L. Giles, Daniel Jonathan Gershman

**Affiliations:** 1Space and Plasma Physics, School of Electrical Engineering and Computer Science, KTH Royal Institute of Technology, 114 28 Stockholm, Sweden; vaivads@kth.se (A.V.); savvra@kth.se (S.R.); pal@kth.se (P.-A.L.); 2NASA Goddard Space Flight Center, Greenbelt, MD 20771, USA; barbara.giles@nasa.gov (B.L.G.); daniel.j.gershman@nasa.gov (D.J.G.)

**Keywords:** space plasma, electron kinetic entropy, quasi-perpendicular shock, adiabatic index

## Abstract

We use Magnetospheric Multiscale (MMS) data to study electron kinetic entropy per particle $S_{e}$ across Earth’s quasi-perpendicular bow shock. We have selected 22 shock crossings covering a wide range of shock conditions. Measured distribution functions are calibrated and corrected for spacecraft potential, secondary electron contamination, lack of measurements at the lowest energies and electron density measurements based on plasma frequency measurements. All crossings display an increase in electron kinetic entropy across the shock $\Delta S_{e}$ being positive or zero within their error margin. There is a strong dependence of $\Delta S_{e}$ on the change in electron temperature, $\Delta T_{e}$, and the upstream electron plasma beta, $\beta_{e}$. Shocks with large $\Delta T_{e}$ have large $\Delta S_{e}$. Shocks with smaller $\beta_{e}$ are associated with larger $\Delta S_{e}$. We use the values of $\Delta S_{e}$, $\Delta T_{e}$ and density change $\Delta n_{e}$ to determine the effective adiabatic index of electrons for each shock crossing. The average effective adiabatic index is $\langle \gamma_{e}\rangle =1.64\pm 0.07$.

## 1. Introduction

Collisionless shock waves are ubiquitous throughout our universe. There are many open questions about the physical mechanisms behind electron heating [1,2,3] and entropy generation [4] at collisionless shocks. In the absence of collisions, dissipation of the solar wind bulk flow energy must be sustained via other processes. The physics behind dissipative processes and entropy generation in collisionless plasmas is an ongoing research topic and numerous studies have been performed observationally, experimentally, theoretically and numerically [5,6,7,8].

The generation of entropy is linked to irreversible dissipation in closed thermodynamic systems [9,10]. However, the utilization of entropy in open thermodynamic systems is currently under debate [6]. For example, Liang et al. [5] use local kinetic entropy density as a diagnostic tool to indicate the dissipation regions in magnetic reconnection events, as seen in numerical simulations. Another study uses the Cluster spacecraft ion and electron data to measure the entropy development across Earth’s quasi-perpendicular bow shock [4]. Both magnetic reconnection events and bow shock crossings are not closed systems. However, the concept of entropy has been successfully used to study the irreversible processes within those systems.

The kinetic entropy is calculated directly from the distribution function *f* according to,
(1)$$s=-k_{B}\int f\ln fd^{3}v.$$

A detailed derivation of Equation (Equation 1) can be found in Liang et al. [5]. We normalize Equation (Equation 1) with the number density $n=\int fd^{3}v$ such that we obtain the entropy per particle
(2)$$S=\frac{s}{n}=-\frac{k_{B}\int f\ln fd^{3}v}{\int fd^{3}v}.$$

In the rest of this paper, when referring to “entropy”, we mean *S*, i.e., entropy per particle.

In the case of the bow shock, theoretical calculations show that entropy generation is localized to the shock transition layer and related to the turbulence collision term, i.e., entropy being generated by wave–particle interactions [11]. There have been few experimental studies of such an entropy development. This is primarily because they require very accurate distribution function calibrations. The Magnetospheric Multiscale (MMS) spacecraft, with its high-resolution particle instrumentation, allows addressing this important topic [12]. Of particular importance is to study how electron kinetic entropy relates to the shock parameters, such as Alfvénic Mach number $M_{A}$, plasma beta $\beta_{e}$, shock angle $\theta_{\mathrm{Bn}}$ and change in density $\Delta n_{e}$ and temperature $\Delta T_{e}$. Earlier studies have shown that there are two critical Whistler Mach numbers, discussed in Krasnoselskikh et al. [13] and Lalti et al. [14]. The Whistler Mach numbers depend on the shock normal angle, $\theta_{\mathrm{Bn}}$, and upstream electron plasma beta, $\beta_{e}$. For Alfvén Mach numbers above these critical values, Whistler waves are not able to form standing wave trains upstream of the shock [15]. Whistler waves can scatter electrons via the cyclotron resonance [16] and could possibly contribute to entropy generation; thus, it is of importance to understand if these two different critical Mach numbers affect the electron entropy development across the shock.

In this study, we use the MMS data to investigate the change in electron kinetic entropy across quasi-perpendicular shocks with different shock parameters, including shocks above and below the critical Whistler Mach numbers. All shocks investigated are supercritical shocks, meaning all have Alfvén Mach numbers above the critical value of 2.76. Above this critical value, anomalous resistivity can no longer dissipate all the incoming bulk flow energy, and the shock starts to reflect incoming ions [17]. In addition, we provide a simplified theoretical estimate (Appendix A) of the change in *total* entropy per particle across a collisionless shock and its dependence on the upstream Alfvénic Mach number, $M_{A}$, shock angle, $\theta_{\mathrm{Bn}}$, and *total* plasma beta, β.

## 2. Method

### 2.1. Data

We used data from 22 shock crossings observed by the MMS spacecraft. All the selected events are during time periods when burst data are available. The shock crossings were chosen to have a wide range of different shock conditions. Some of the crossings were selected from the databases used in Raptis et al. [18] and Lalti et al. [14]. Magnetic field data were taken from the fluxgate magnetometer (FGM) with a resolution of 7.8 ms [19]. All plasma moments and the measured 3D distribution functions were obtained from the fast plasma investigation (FPI) measuring at a cadence of 30 ms [20]. The electric field spectra and spacecraft potential were obtained from the spin plane double probe (SDP) [21] and the axial double probe (ADP) [22] instruments. The solar wind ion temperature was not measured accurately by MMS; therefore, it was obtained from the 1-min resolution OMNI database [23].

### 2.2. Shock Parameter Calculations

#### 2.2.1. Shock Normal

The shock normal direction was determined using upstream and downstream measurements of the magnetic field and ion velocity at MMS1. The normal vector was calculated using four different methods: the *velocity coplanarity method* and three *mixed methods* [24]. The sign of the normal vector is taken such that the normal points upstream from the shock. The shock normals listed in Table 1 are calculated using the average of the three mixed methods. However, one crossing exhibited an inconsistency between the observations and calculated $\theta_{\mathrm{Bn}}$ for all methods except the velocity coplanarity method. Hence, the velocity coplanarity method was used for this crossing.

#### 2.2.2. Upstream Shock Parameters

We characterize the shock by the following upstream parameters: the plasma beta β, the shock normal angle $\theta_{\mathrm{Bn}}$, the Alfvén Mach number $M_{A}$, the fast magnetosonic Mach number $M_{\mathrm{ms}}$, the linear whistler Mach number $M_{\mathrm{wh}}$, and the non-linear whistler Mach number $M_{\mathrm{whn}}$. Below we describe each of these parameters.

The plasma beta is the ratio between the particle pressure and the magnetic pressure
(3)$$\beta_{j}=\frac{n_{j}k_{B}T_{j}}{B^{2}/2\mu_{0}}$$
where *j* indexes each plasma species. The shock normal angle $\theta_{\mathrm{Bn}}$ is the angle between the upstream magnetic field $B_{u}$ and the shock normal $\hat{n}$,
(4)$$\theta_{\mathrm{Bn}}=\arccos\left(\frac{B_{u}\cdot \hat{n}}{|B_{u}|}\right).$$
Using Equation (Equation 4), we obtain three $\theta_{\mathrm{Bn}}$ values from the three different shock normals obtained via the three mixed methods. The $\theta_{\mathrm{Bn}}$ values given in Table 2 are the average of these three values.

In the theory of collisionless shocks, there are different Mach numbers governing the physics. The Alfvén Mach number $M_{A}$ is the ratio of the normal component of the upstream flow velocity relative to the shock to the upstream Alfvén speed, $V_{A}$. The fast magnetosonic Mach number $M_{\mathrm{ms}}$ is calculated from the Alfvén Mach number, total plasma beta and shock angle according to
(5)$$M_{\mathrm{ms}}^{2}=\frac{2M_{A}^{2}}{1+5\beta /6+{\left[{\left(1-5\beta /6\right)}^{2}+\left(10\beta /3\right)\sin^{2}\theta_{\mathrm{Bn}}\right]}^{1/2}},$$
where $\beta =\beta_{e}+\beta_{i}$ is the upstream total plasma beta. In this study, we also consider two critical whistler Mach numbers. The linear whistler Mach number $M_{\mathrm{wh}}$, defined as the ratio of maximum whistler phase speed and the normal component of the upstream flow velocity relative to the shock, can be expressed as by Oka et al. [15],
(6)$$M_{\mathrm{wh}}=\frac{1}{2}\sqrt{\frac{m_{i}}{m_{e}}}\left|\cos\theta_{\mathrm{Bn}}\right|,$$
which depends only on the shock normal angle. For $M_{A}>M_{\mathrm{wh}}$, the whistler wave phase velocity cannot be directed upstream away from the shock and thus allow the formation of standing wave fronts in front of quasi-perpendicular shocks. A similar expression for the linear whistler Mach number can be defined with respect to the group velocity instead, see, e.g., Oka et al. [15]. Krasnoselskikh et al. [13] also introduce a non-linear whistler Mach number ($M_{\mathrm{whn}}$), satisfying the equation
(7)$$M_{\mathrm{whn}}^{2}{\left(2M_{\mathrm{wh}}^{2}-M_{\mathrm{whn}}^{2}\right)}^{3}=\beta_{e}{\left(2M_{\mathrm{whn}}^{2}-M_{\mathrm{wh}}^{2}\right)}^{3},$$
that allows finding $M_{\mathrm{whn}}$ numerically. Unlike the linear whistler Mach number in (Equation 6), the non-linear whistler Mach number not only depends on the shock angle but also has a weak dependence on the electron plasma beta, $\beta_{e}$. For $M_{A}>M_{\mathrm{whn}}$, non-linear whistler wave trains are not able to form standing wave fronts upstream of the quasi-perpendicular shock [13].

Calculating the Mach numbers associated with a shock wave, the normal component of the upstream flow velocity relative to the shock needs to be known. It is calculated from the measured solar wind velocity $V_{\mathrm{SW}}$ (see Table 1), shock normal, and the shock velocity. The shock velocity is determined using two methods: *the mass flux method* and *the Smith and Burton method* [24]. The shock velocity in our calculations is taken as the average between the two methods. However, if one of the methods yields an unphysical velocity, such as the shock velocity having a sign inconsistent with observations, then the shock velocity from the other method is used. All the calculated Mach numbers are given in Table 2 and Table 3.

### 2.3. Electron Distribution Function Corrections

To make reliable estimates of the electron entropy and electron moments, the electron distribution functions from the MMS data archive have to be additionally calibrated. Four corrections are performed, all of them described below.

#### 2.3.1. Spacecraft Potential

The spacecraft has a potential Φ with respect to the ambient plasma environment. Hence, a charged particle with energy *E* will be measured by MMS at a shifted energy
$$E^{\prime }=E-q\Phi ,$$
where *q* is the charge of the particle. Normally, MMS spacecraft are positively charged, and therefore, electrons are accelerated by the spacecraft and measured at higher energies, while (positively charged) ions are decelerated and measured at lower energies. We correct the measured distribution functions of ions and electrons by using the spacecraft potential estimates from the electric field instrument. This is the only correction performed on the ion distribution functions measured by MMS.

#### 2.3.2. Secondary Electrons

Secondary electron emissions contaminate the lower energy channels of the electron distribution functions measured by MMS [25]. This contamination is illustrated in Figure 1a, which shows the electron distribution function average over all angles. The effect of the secondary electrons can be seen at the lower energies, E<20eV, as a significant increase in the distribution function. The secondary electron number density increases with increasing background plasma density. This is attributed to secondary electron emissions due to background plasma electrons hitting the spacecraft and instrument surfaces.

In order to resolve the distribution function of the ambient background plasma, the low-energy secondary electron population needs to be removed. This is achieved by modeling the secondary electron population as consisting of two components,
(8)$$f_{\sec}=f_{\sec,\mathrm{ph}}+f_{\sec,\mathrm{iso}},$$
one sun angle-dependent population due to photoelectrons ($f_{\sec,\mathrm{ph}}$) and one isotropic population of secondary electron emission created by the impact of plasma electrons ($f_{\sec,\mathrm{iso}}$). The sun angle-dependent component is independent of the background ambient plasma density and is set according to Gershman et al. [25], corresponding to an efficient photoelectron density of $n_{\sec,\mathrm{ph}}=0.35\;{\mathrm{cm}}^{-3}$. The isotropic component is set to model the secondary electron population dependence on the ambient background plasma density. This model requires knowledge about the efficient number density of secondaries. This number density is estimated using combinations of the values at the two lowest energy channels. We define the partial density $n(E_{i})$ to be the number density of electrons corresponding to the distribution function of secondaries with energies between $E_{i}-\Delta E_{i}$ and $E_{i}+\Delta E_{i}$, where *i* is the index of each energy bin of the electron instrument and $\Delta E_{i}$ is the corresponding half of the energy bin size. Figure 2 shows partial densities, $n(E_{i})$, at four different locations during a shock transition. The secondary electron model distribution function is then subtracted from the one measured by MMS according to
(9)$$f_{\mathrm{corr}}=f_{\mathrm{MMS}}-f_{\sec}.$$

The secondary electron distribution function, $f_{\sec}$, is adjusted so that $f_{\mathrm{corr}}$ resembles a Maxwellian-like shape at the low energies in the solar wind. In the magnetosheath, $f_{\sec}$ is adjusted to yield $f_{\mathrm{corr}}$ as a flat-top distribution. Panels (a) and (b) of Figure 1 show the measured distribution functions, $f_{\mathrm{MMS}}$, and the corrected distribution functions, $f_{\mathrm{corr}}$, at five different shock crossing locations indicated in panel (c).

For electron distribution functions in the shock transition layer (STL), secondary electron density is estimated by performing a gradual linear transition between the secondary densities used in the solar wind and in the magnetosheath.

#### 2.3.3. Extrapolation to Zero Energy

The part of the distribution functions ranging from the lowest measured energy bin value down to zero energy, while not measured by the spacecraft, gives an important contribution to the entropy estimates. Therefore, it is important to use good approximations for the distribution function values in that interval. We assume that in the low energy range, the distribution of electrons is isotropic, and any angular dependence is assumed negligible. This is a good approximation as the electron drift velocity is small compared to their thermal velocity. Based on these assumptions, we extrapolate the lowest value of (the solid angle averaged) distribution functions down to zero. This extrapolation is illustrated in Figure 3 as a dashed line. The entropy calculation of this extrapolated part is shown below in Section 2.4.

#### 2.3.4. Density and Temperature

As a final step, the corrected distribution functions are scaled by a scalar factor η such that they correspond to the same density as obtained based on the measurements of plasma frequency by the electric field instrument upstream and ion density downstream. As an example, Figure 4 shows the electric field spectrogram for one shock crossing. The plasma frequency peak is seen in the upstream solar wind at roughly 30 kHz, corresponding to a plasma density of about $11.3\;{\mathrm{cm}}^{-3}$. We verify the plasma frequency peak by comparing the derived density with the measured one. Furthermore, the plasma frequency is much higher than the gyrofrequency (around 200 Hz). Therefore, the upper hybrid frequency is approximately equal to the plasma frequency. In this case, all the corrections from Section 2.3.1, Section 2.3.2 and Section 2.3.3 provide a distribution function corresponding to a density of about $9.44\;{\mathrm{cm}}^{-3}$, and thus, the scaling factor is η=11.3/9.44=1.2. We use this scaling factor for all distribution functions in the upstream solar wind, including up to the beginning of the shock ramp. Figure 5a displays the measured electron and ion densities as supplied by the MMS data center and the corrected density (red line) that we use in our study.

Figure 5b shows the electron temperature values as supplied by the MMS data center (blue) and obtained from our corrected distribution function (red). The slight differences between the values in the magnetosheath are most probably due to different handling of secondary electrons, and in our study, we use temperature values that we obtain from the corrected distribution functions.

### 2.4. Electron Entropy Calculation

The measured electron entropy density is calculated as
(10)$$s=-k_{B}\int f\ln\left(f\right)d^{3}v=s_{0}-k_{B}\sum_{i,j,k}f_{ijk}\ln\left(f_{ijk}\right)\Delta v_{ijk}$$
where $f_{ijk}$ is the corrected distribution function measured by the electron instrument at a specific energy channel $E_{i}$ having velocity space volume $\Delta v_{ijk}$ defined by the energy bin size $\Delta E_{i}$ and spherical angular bin sizes $\Delta \theta_{j}$ and $\Delta \varphi_{k}$,
(11)$$\Delta v_{ijk}\propto \sqrt{\left(E_{i}-e\Phi \right)}\Delta E_{i}\sin\theta_{j}\Delta \theta_{j}\Delta \varphi_{k},$$
and $s_{0}$ is the entropy calculated from the extrapolated part of the distribution function (illustrated by the horizontal dashed lines in Figure 3). We are assuming that the distribution function is constant at these energies, and therefore, $s_{0}$ is obtained as
(12)$$s_{0}=-k_{B}\int_{0}^{2\pi }\int_{0}^{\pi }\int_{0}^{E_{0}}f\ln fd^{3}v=-k_{B}f_{0}\ln f_{0}\frac{8\pi E_{0}^{3/2}}{3}$$
where $E_{0}$ denotes the lower edge of the lowest electron energy bin value (5.51eV−Φ), and $f_{0}$ is the value of the distribution function used to extrapolate down to zero energy. Finally, the entropy per particle is calculated according to Equation (Equation 2).

## 3. Results

Figure 6 displays one example shock event, henceforth referred to as crossing 5, illustrating the analysis performed on each shock crossing. The magnetic field data in Figure 6a shows the characteristic signs of a quasi-perpendicular shock crossing. At the start of the interval, the spacecraft are upstream of the shock with an average solar wind speed of about 439 km/s, see Figure 6b. Around 07:35:27 UTC, the ion velocity starts to decrease and the spacecraft enters the foot region of the shock. The ion spectrogram in Figure 6c shows an almost monoenergetic signal at about 1 keV energy in the upstream region that corresponds to the cold solar wind beam. There is a high energy ion population with energy up to 10 keV, associated with the shock, present already at 07:35:21 UTC. Hence, the solar wind upstream parameters are taken as averaged values before this time. At around 07:35:36 UTC, the MMS1 spacecraft measures a sharp increase in density and temperature, see Figure 6e,f. This increase coincides with a sharp transition in the ion (Figure 6c) and electron (Figure 6d) differential energy flux and is interpreted as the shock ramp. The sharp peak seen in the magnetic field, density and temperature at around 07:35:38 UTC is identified as the overshoot, which is followed by the downstream magnetosheath. The calculated entropy of electrons and ions (panels (g) and (h)) is observed to increase across the shock. We define upstream and downstream parameters by taking 6-s averages, 07:35:15–07:35:21 UTC upstream and 07:35:54–07:36:00 UTC downstream. The Alfvén Mach number of this shock is $M_{A}\approx 10.9$ and the shock angle is $\theta_{\mathrm{Bn}}\approx 61^{\circ }$. The change in electron entropy across the shock is $0.39k_{B}$, and the change in ion entropy is 2.9$k_{B}$. All the other shocks have been analyzed in a similar manner.

Table 1, Table 2 and Table 3 show the shock parameters for the 22 analyzed crossings. The number, date and time of each shock are shown in Table 1, and reference to a specific shock crossing is hereafter made using its number. The averaged upstream values of the magnetic field, ion velocity, electron density, electron temperature and the shock normal in GSE-coordinates for each crossing are also included in Table 1. Table 2 and Table 3 show the change in electron entropy, $\Delta S_{e}$, along with 10 other shock parameters, including the fast magnetosonic Mach number, Alfvén Mach number, upstream electron plasma beta, shock angle, change in density, change in electron temperature, adiabatic index, linear and non-linear whistler Mach numbers and maximum electric field strength measured at the shock. The effective adiabatic index for electrons is calculated based on an analytical expression; under the assumption of constant heat capacities and reversibility, the change in entropy can be expressed as [9,17]
(13)$$\Delta S=k_{B}\ln\left[\frac{n_{\mathrm{SW}}}{n_{\mathrm{MSh}}}{\left(\frac{T_{\mathrm{MSh}}}{T_{\mathrm{SW}}}\right)}^{\frac{1}{\gamma -1}}\right]$$
Assuming (Equation 13) applies for the different plasma species separately, the effective adiabatic index $\gamma_{e}$ for electrons can be determined for each shock crossing (see last column in Table 2). It should be noted that Equation (Equation 13) is derived under assumptions not necessarily valid for a collisionless shock. However, modeling the shock this way allows us to determine the adiabatic indices from quantities derived via our corrected distribution function and compare our results to previous work by Schwartz et al. [26] and Pudovkin et al. [27] (see following section for further discussion). With 22 shock crossings analyzed, a statistical study can be performed. In Figure 7, the change in kinetic electron entropy per particle, $\Delta S_{e}$, is plotted against the parameters listed above. Figure 8 depicts the effective adiabatic indices for each shock crossing (panel (a)) and agreement with Equation (Equation 13) (panel (b)).

## 4. Discussion

According to Table 2, the kinetic entropy increases across the shock in nearly all of the crossings, except two that are discussed in the next paragraph. With energy being dissipated at the shock, an increase in entropy is expected [17]. Collisionless shocks generating entropy have been reported by Parks et al. [4] based on Cluster data. The entropy increase found by Parks et al. [4], $\Delta S_{e}\approx 2k_{B}$, is significantly larger than the entropy increases found in this study, from $-0.06k_{B}$ to $1.4k_{B}$. One reason for this inconsistency can be the different definition of entropy used in Parks et al. [4] where the entropy is calculated using the Gibbs entropy formulation $S=-k_{B}\sum p_{j}\ln p_{j}$ and *j* indexes the sampled phase space volume element, $p_{j}=f_{j}\Delta^{3}v_{j}/n$. This definition of entropy is very sensitive to the instrument resolution of the spacecraft, while calculating the entropy directly from the distribution function allows comparisons to theory and simulations [5,6].

Although this paper focuses solely on electrons, the ion kinetic entropy is shown for completeness in Figure 6h. Ion entropy generation at collisionless shocks is an important topic of its own. However, the solar wind ion distribution function is not resolved properly for the MMS satellites. Hence, the calculated ion entropy in the solar wind can not be fully trusted. Therefore, a study of ion kinetic entropy is left for future studies.

Crossings 2 and 18 show a $\Delta S_{e}\approx -0.06<0$, i.e., the entropy seems to decrease across the shock. Note that both crossings exhibit $\Delta S_{e}\geq 0$ values within their statistical error margin (see Table 2) and the change is rather close to zero. Thus, the small negative value might be due to statistical fluctuations. Alternatively, it can be due to leftover secondary electrons from the manual removal procedure described in Section 2.3.2. The secondary electron contamination has a decreasing effect on the change in electron entropy, $\Delta S_{e}$. If the real change in entropy is positive but close to zero, any small leftover part of secondary electrons could make the calculated $\Delta S_{e}$ negative.

The gradual linear transition of the efficient secondary density used for the STL is made mostly for illustration purposes, and drawing conclusions about entropy changes within the STL must be taken with some consideration.

The standard deviations (errors) presented for $\Delta n_{e}$, $\Delta T_{e}$, $\Delta S_{e}$ and the adiabatic index should be viewed as a minimum error. There can be a systematic error introduced by the secondary electron removal procedure that is not accounted for in the standard deviations presented in Table 2 and Figure 8. This systematic error can originate from the manual fit of the distribution function described in Section 2.3.2. An additional possible source of error can be that no measurements of the plasma frequency in the magnetosheath downstream of the shocks are available to cross-check the magnetosheath densities. However, we expect the systematic error in the magnetosheath to be small as both ions and electrons have high thermal energies allowing good measurements, and particle instruments are optimized for measuring magnetosheath plasma.

Figure 7 shows the correlation of the shock parameters with the entropy change $\Delta S_{e}$. Two parameters have a clear correlation with the entropy changes. First, Figure 7c shows a clear relationship between $\Delta S_{e}$ and $\Delta T_{e}$. This is expected due to energy dissipating processes occurring at the shock [17]. Irreversible processes always increase entropy and temperature [10]; hence, we expect an increase in entropy to yield an increase in temperature. Secondly, Figure 7e shows a relationship between $\Delta S_{e}$ and the electron plasma beta, $\beta_{e}$. High $\Delta S_{e}$ is associated with low solar wind electron plasma beta, $\beta_{e}<1$, while low $\Delta S_{e}$ is associated with high solar wind electron plasma beta, $\beta_{e}>1$. This inverse electron beta-dependence is qualitatively similar to what is theoretically predicted for the *total* entropy per particle change vs. the *total* plasma beta across a collisionless shock assuming a one-fluid MHD approximation, see Appendix A and Figure A2. However, it is not obvious why similar relation should hold only for electron beta-dependence. Future work can address this point by considering a two-fluid plasma description. For the other shock parameters, no clear correlation with $\Delta S_{e}$ is observed. Due to the strong $\Delta S_{e}$-dependence on the electron plasma beta, we suggest that in future work, the dependence on other parameters can be analyzed using more crossings but in a limited range of $\beta_{e}$.

Now, we look at the effective adiabatic index for electrons $\langle \gamma_{e}\rangle$. Based on Figure 8a, we can estimate $\langle \gamma_{e}\rangle$ for each of the crossings using Equation (Equation 12). We obtain that the effective adiabatic index of $\langle \gamma_{e}\rangle =1.64\pm 0.07$. This value is in the vicinity of γ≈5/3, suggesting the electrons behave similarly to a monatomic ideal gas with three degrees of freedom. This is consistent with previous work by Pudovkin et al. [27] but inconsistent with previous work by Schwartz et al. [26]. Schwartz et al. [26] use data from several different planetary bow shock encounters in the heliosphere and determine an effective polytropic index using upstream and downstream measurements of the electron pressure and density. Their study reports ideal gas behavior only for subcritical shocks and one-dimensional ($\gamma_{\mathrm{eff}}\approx 3.0$) behavior for supercritical shocks. All our shocks are supercritical and still show γ≈5/3. The discrepancy might be explained by the lower quality of data used in [26] and/or physical differences between heliospheric shocks.

We also compare our entropy values of the observed distribution functions with the maximum entropy values calculated using the Maxwellian distribution having the same electron density and temperature as the observed distribution function. Given plasma density *n* and temperature *T*, Maxwellian distribution is defined
(14)$$f\left(v\right)=n{\left(\frac{m}{2\pi k_{B}T}\right)}^{3/2}e^{-mv^{2}/2k_{B}T}$$
and using the definition of kinetic entropy density in Equation (Equation 1), an analytical expression can be found for a maximum entropy [6]:(15)$$S_{M}=\frac{3}{2}k_{B}\left[1+\ln\left(\frac{2\pi k_{B}T}{mn^{2/3}}\right)\right].$$

Figure 9 shows a comparison between the entropy of a Maxwellian distribution and the entropy calculated using the corrected distribution functions. The calculated entropy is strictly less than the maximum state (Maxwellian) throughout the interval. This is expected as distribution functions are not close to being Maxwellian.

## 5. Conclusions

We use MMS data to calculate the kinetic entropy per particle across Earth’s quasi-perpendicular bow shock. All data are obtained using the MMS1 spacecraft. With the close separation of the spacecraft and the change in entropy obtained by taking upstream and downstream averages, the difference between the spacecraft data will be negligible. Altogether 22 quasi-perpendicular shock crossings have been analyzed. The electron kinetic entropy per particle is calculated using the kinetic definition of entropy and the distribution function measured by MMS1. It is shown that the measured electron distribution function needs further calibrations when calculating the electron entropy, density and temperature. The calibrations include the corrections for the spacecraft potential, the removal of secondary electron emissions, the extrapolation of distribution function values to zero energy and the density calibration using the observed plasma frequency emissions. Our main findings are:In total, 20 out of 22 crossings display an increase in the electron kinetic entropy going from the solar wind to the magnetosheath in the range $\Delta S_{e}\approx 0.1-1.4k_{B}$.Two crossings display a slight decrease, $\Delta S_{e}/k_{B}\approx -0.06\pm 0.06$ and $\Delta S_{e}/k_{B}\approx -0.06\pm 0.09$, but within error margins, they are still consistent with entropy not decreasing across the shock.We observe that $\Delta S_{e}$ displays a strong dependence on the change in electron temperature across the shock, $\Delta T_{e}$, and the upstream electron plasma beta, $\beta_{e}$. Shocks with high $\Delta T_{e}$ are found to have high $\Delta S_{e}$. Shocks with small upstream $\beta_{e}$ are found to generate more entropy than shocks with large upstream $\beta_{e}$.For the parameters, $M_{A}$, $M_{\mathrm{wh}}$, $M_{\mathrm{whn}}$, $T_{\mathrm{SW}}$, $\Delta n_{e}$ and $E_{\max}$, no clear trend is observed, and more crossings need to be analyzed.The effective adiabatic index of electrons is calculated for each shock crossing, using the analytical expression relating entropy change to the change in density and temperature, see Equation (Equation 13). We find that all shocks have an effective adiabatic index $\langle \gamma_{e}\rangle =1.64\pm 0.07$ that is in the vicinity of 5/3. This suggests the electrons behave similarly to a monatomic gas with three degrees of freedom.

## Figures and Tables

**Figure 1 entropy-24-00745-f001:**
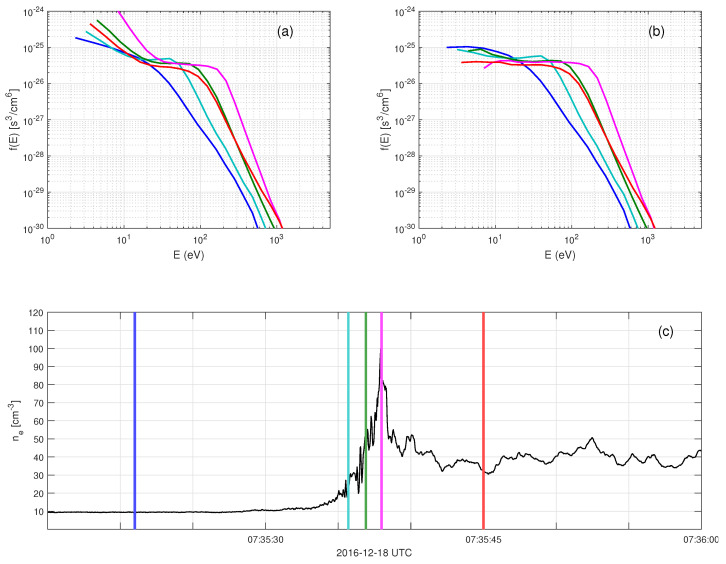
The correction of secondary electron contamination at the low energies. (**a**) Measured distribution function $f_{\mathrm{MMS}}$, averaged over a full solid angle, (**b**) corrected distribution function $f_{\mathrm{corr}}$, (**c**) electron number density. The times of distribution functions are indicated by the colored vertical lines in panel (**c**). The different colors correspond to different times crossing the shock.

**Figure 2 entropy-24-00745-f002:**
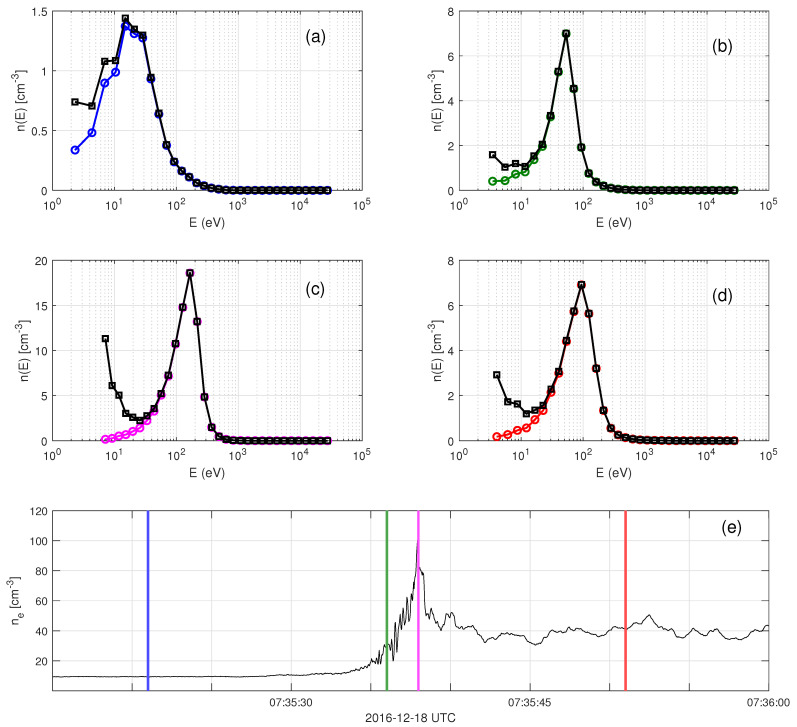
The correction of secondary electron contamination as seen in partial densities for MMS1. (**a**–**d**) Partial densities at four different locations crossing the shock. The black lines are calculated using the measured $f_{\mathrm{MMS}}$ and the colored lines are calculated using the corrected $f_{\mathrm{corr}}$. (**e**) The electron density. The colored vertical lines indicate the time instants of $f_{\mathrm{MMS}}$ measurements. The different colors correspond to different times crossing the shock. The number density of secondary electrons is estimated using combinations at the values of the two lowest energy channels.

**Figure 3 entropy-24-00745-f003:**
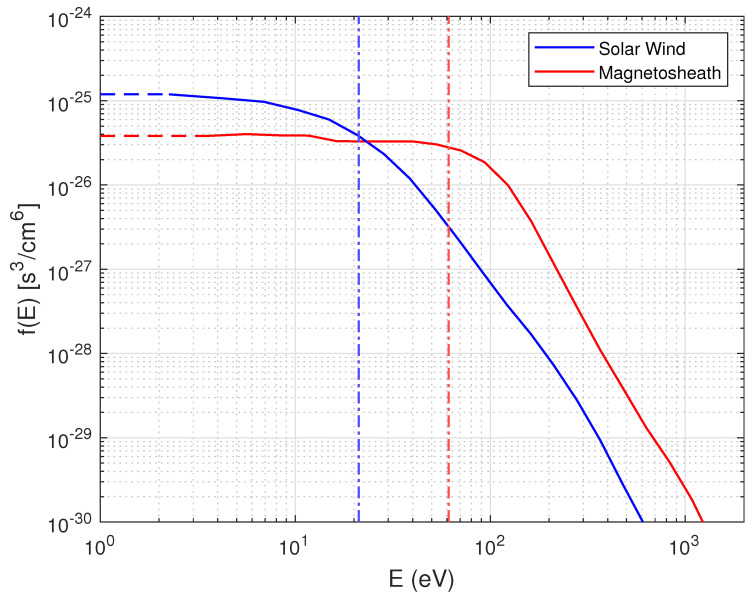
Distribution functions in upstream solar wind (blue) and downstream magnetosheath (red). The horizontal dashed lines indicate the extrapolated part of the distribution functions, while the vertical dashed-dotted lines indicate the thermal energies of the entire distribution functions.

**Figure 4 entropy-24-00745-f004:**
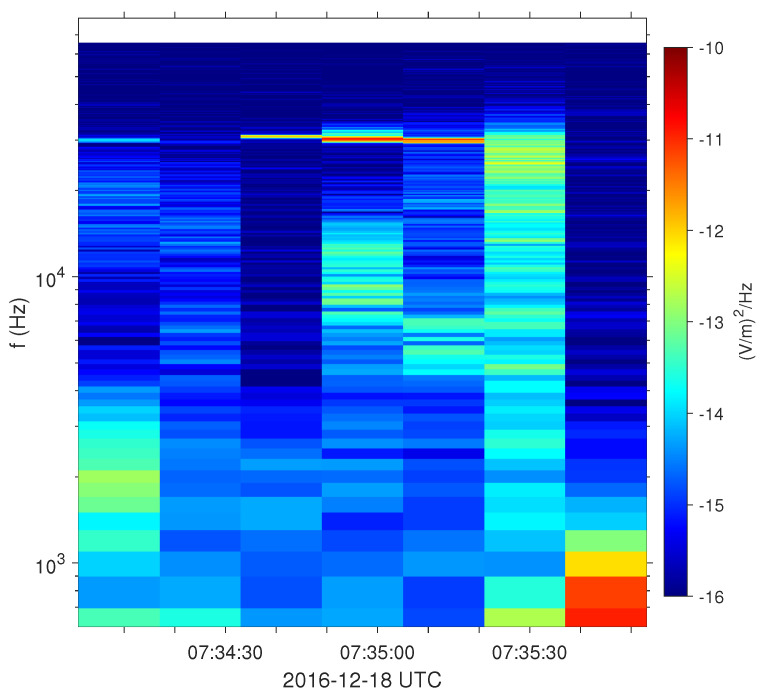
Electric field spectrogram for one shock crossing. The plasma frequency emission at f≈30.1kHz corresponds to a density of about $11.3\;{\mathrm{cm}}^{-3}$.

**Figure 5 entropy-24-00745-f005:**
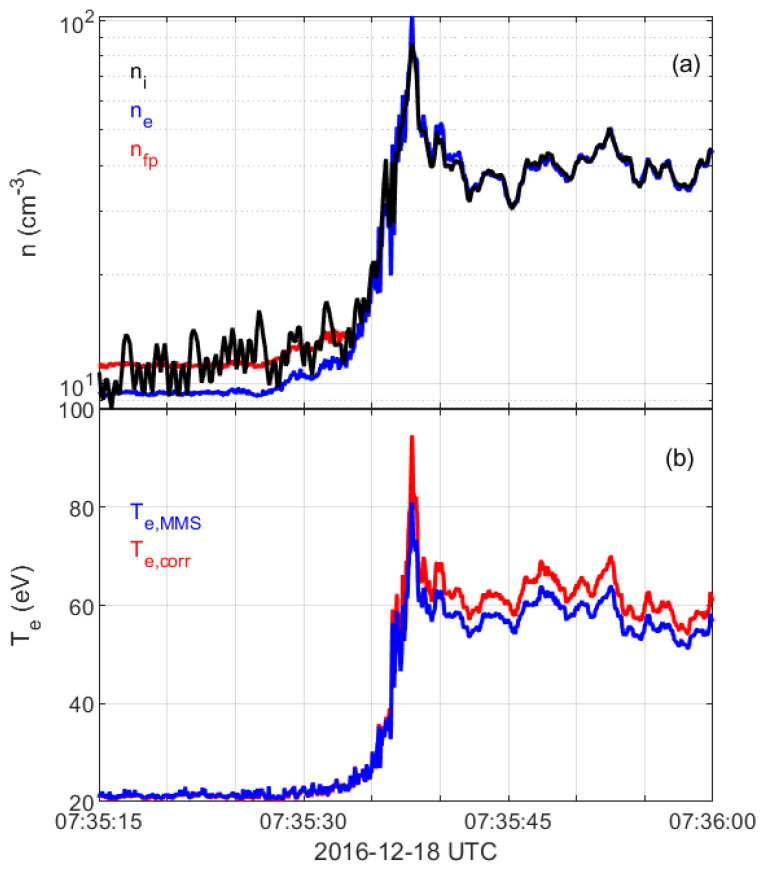
Density and temperature across the shock. (**a**) Density as obtained by the ion instrument (black), electron instrument (blue) and density obtained from the corrected distribution function using plasma frequency correction (red). (**b**) Electron temperature as given from the data center (blue) and calculated using the corrected distribution function (red).

**Figure 6 entropy-24-00745-f006:**
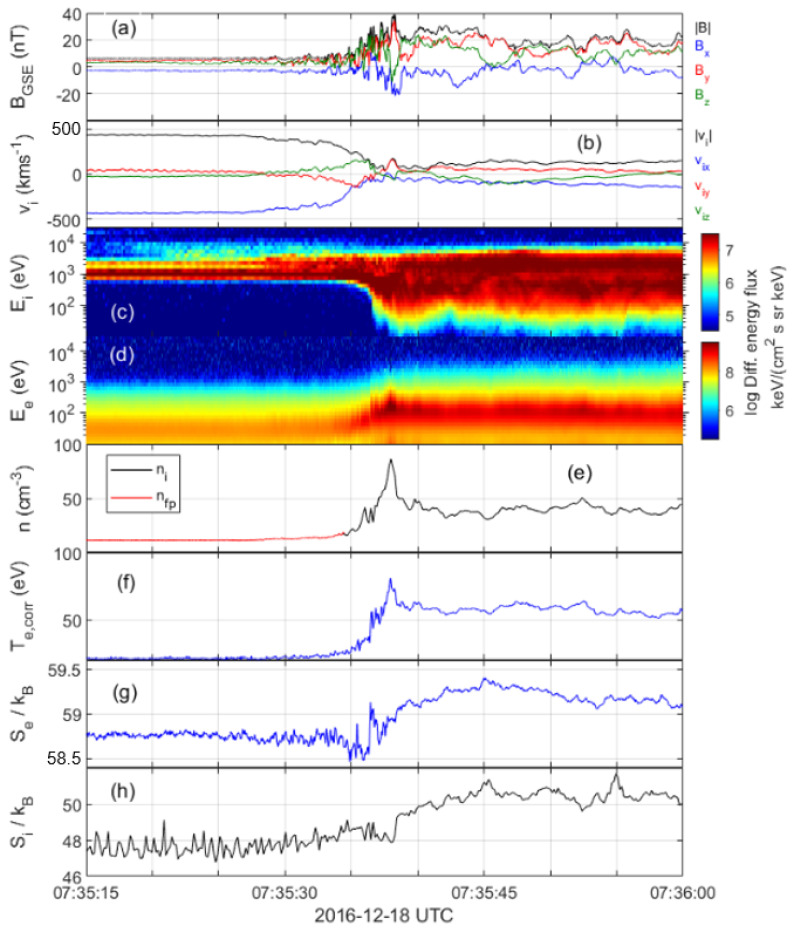
MMS1 measurements of shock crossing 5. The panels show, (**a**) magnetic field, (**b**) ion velocity, (**c**) ion spectrogram, (**d**) electron spectrogram, (**e**) measured ion density (black) and density obtained from plasma frequency (red), (**f**) calculated electron temperature, (**g**) electron kinetic entropy per particle and (**h**) ion kinetic entropy per particle.

**Figure 7 entropy-24-00745-f007:**
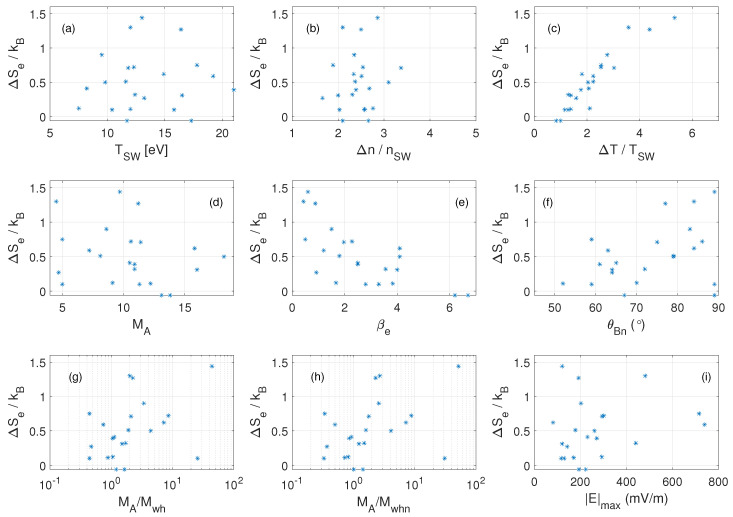
The change in kinetic electron entropy per particle ($\Delta S_{e}$) plotted against (**a**) solar wind electron temperature, (**b**) difference in electron number density, (**c**) difference in electron temperature, (**d**) Alfvénic Mach number, (**e**) upstream electron plasma beta, (**f**) shock normal angle, (**g**) Alfvénic to linear whistler Mach number ratio, (**h**) Alfvénic to non-linear whistler Mach number ratio and (**i**) maximum electric field strength measured across the shocks. Every asterisk represent a shock crossing.

**Figure 8 entropy-24-00745-f008:**
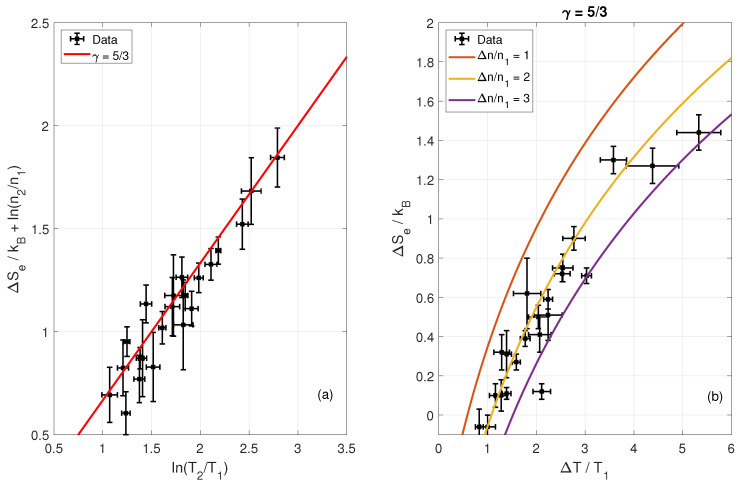
The effective (average) adiabatic index for electrons of all 22 shock crossings is calculated to be $\langle \gamma_{e}\rangle =1.64\pm 0.07$. Panel (**a**) shows the agreement with an adiabatic index of γ=5/3, which is the expected value for a monatomic gas with 3 degrees of freedom. Panel (**b**) displays the change in entropy vs. change in temperature (black asterisks) and the analytical expression in Equation (Equation 13) for three different density ratios with $\gamma_{e}=5/3$. Here, index 1 indicates solar wind and 2 magnetosheaths.

**Figure 9 entropy-24-00745-f009:**
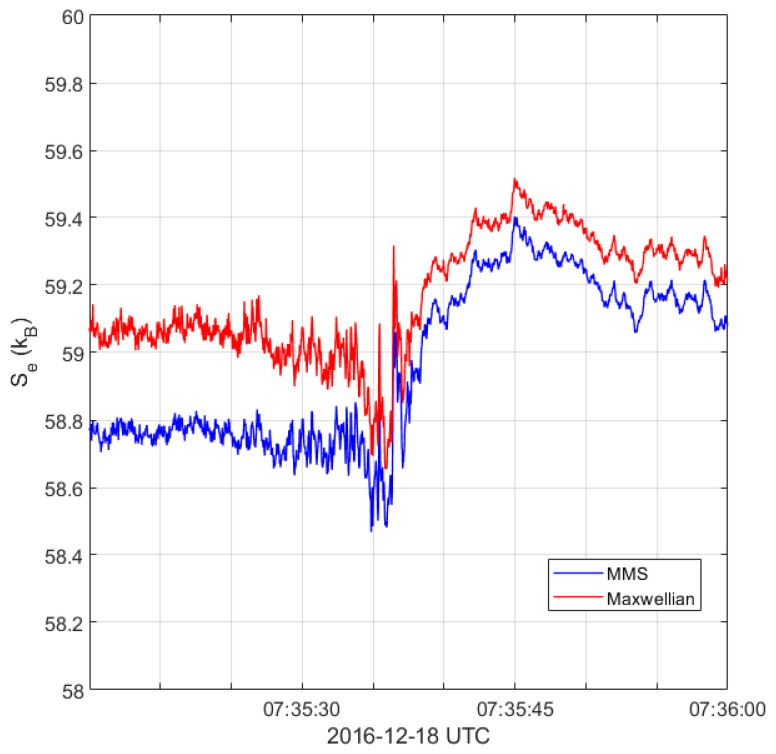
Comparison of the calculated entropy (blue) to the entropy of a Maxwellian distribution (red) obtained from Equation (Equation 15). The temperature and density in Equation (Equation 15) are evaluated at each point in the shock interval and plotted as the red lines in Figure 5.

**Table 1 entropy-24-00745-t001:** The studied shock crossings along with upstream parameters and calculated shock normal vectors. The upstream parameters were obtained taking 6-s averages. The shock normals are obtained as the average of the three mixed methods described in Paschmann and Daly [24] except for crossing 10, where the velocity coplanarity method was used.

Crossing	$B_{\mathrm{SW}}\;\left(\mathrm{nT}\right)$	$V_{\mathrm{SW}}\;\left(\mathrm{km}/s\right)$	$n_{e,\mathrm{SW}}\;\left({\mathrm{cm}}^{-3}\right)$	$T_{e,\mathrm{SW}}\;\left(\mathrm{eV}\right)$	$\hat{n}$ (GSE)
**1**. 2016-11-10 17:10	12.4	379	24.1	19.2	0.88 0.44 0.18
**2**. 2016-11-10 16:59	7.0	373	43.8	17.3	0.81 0.58 0.07
**3**. 2016-12-06 10:55	8.0	343	24.8	11.6	0.97 0.24 0.02
**4**. 2016-12-09 10:29	7.9	617	6.9	13.0	0.99 0.15 −0.06
**5**. 2016-12-18 07:36	6.2	439	11.3	21.0	0.99 −0.09 0.03
**6**. 2017-01-01 09:11	3.9	486	9.2	16.5	0.99 −0.13 0.07
**7**. 2017-01-15 06:43	5.0	324	19.1	10.4	0.88 0.26 0.39
**8**. 2017-01-18 05:39	17.1	374	21.7	17.8	0.95 −0.32 0.003
**9**. 2017-01-31 10:07	9.1	645	11.0	16.4	0.94 −0.24 0.24
**10**. 2017-10-18 04:34	3.3	404	4.7	15.8	0.80 0.55 0.25
**11**. 2017-11-02 04:27	9.9	317	17.0	13.2	0.73 0.66 0.15
**12**. 2017-11-24 23:20	9.1	396	7.3	12.0	0.86 0.48 0.20
**13**. 2017-11-28 18:01	5.2	405	11.3	11.8	0.99 0.03 0.05
**14**. 2017-12-26 22:10	3.2	460	7.1	14.9	0.94 −0.07 0.33
**15**. 2018-01-24 04:05	2.4	369	5.9	9.8	0.92 −0.35 0.16
**16**. 2018-11-16 00:11	4.3	361	7.5	9.5	0.80 0.57 0.18
**17**. 2018-11-18 17:47	5.6	310	17.6	7.5	0.84 0.43 0.32
**18**. 2018-11-27 04:18	3.4	299	16.3	11.7	0.95 0.22 0.24
**19**. 2018-12-16 20:16	4.0	325	11.5	12.4	0.96 0.14 0.25
**20**. 2018-12-25 07:56	4.6	325	9.9	12.3	0.98 0.16 0.11
**21**. 2019-12-17 21:44	3.5	327	9.6	12.0	0.89 0.15 0.42
**22**. 2020-04-17 18:19	3.4	297	8.8	8.2	0.63 −0.78 −0.04

**Table 2 entropy-24-00745-t002:** The studied shock crossings along with the calculated fast magnetosonic Mach number, $M_{\mathrm{ms}}$, Alfvénic Mach number, $M_{A}$, upstream electron plasma beta, $\beta_{e}$, shock angle, $\theta_{\mathrm{Bn}}$, change in density, $\Delta n/n_{\mathrm{SW}}$, change in temperature, $\Delta T_{e}/T_{e,\mathrm{SW}}$, change in electron kinetic entropy per particle, $\Delta S_{e}$ and adiabatic index, $\gamma_{e}$.

#	$M_{\mathrm{ms}}$	$M_{A}$	$\beta_{e}$	$\theta_{\mathrm{Bn}}$	$\Delta n/n_{\mathrm{SW}}$	$\Delta T_{e}/T_{e,\mathrm{SW}}$	$\Delta S_{e}/k_{B}$	$\gamma_{e}$
**1**	6.0	7.1	1.2	63	2.5 ± 0.1	2.2 ± 0.1	0.59 ± 0.05	1.64 ± 0.03
**2**	4.9	13.6	6.2	89	2.1 ± 0.4	1.0 ± 0.2	−0.06 ± 0.06	1.65 ± 0.11
**3**	5.2	7.9	1.8	79	2.4 ± 0.5	2.2 ± 0.3	0.51 ± 0.13	1.68 ± 0.09
**4**	6.0	9.9	0.6	85	2.9 ± 0.4	5.3 ± 0.5	1.44 ± 0.09	1.66 ± 0.04
**5**	5.4	10.9	2.5	61	2.4 ± 0.2	1.8 ± 0.1	0.39 ± 0.04	1.63 ± 0.04
**6**	5.1	14.7	4.0	64	2.0 ± 0.4	1.4 ± 0.1	0.31 ± 0.12	1.62 ± 0.09
**7**	5.2	10.5	3.3	89	2.0 ± 0.2	1.3 ± 0.1	0.10 ± 0.08	1.68 ± 0.09
**8**	4.2	4.8	0.5	59	1.9 ± 0.5	2.5 ± 0.2	0.75 ± 0.07	1.70 ± 0.05
**9**	5.4	10.8	0.9	77	2.5 ± 0.2	4.4 ± 0.5	1.27 ± 0.09	1.67 ± 0.06
**10**	4.2	7.1	2.8	59	2.6 ± 0.3	1.2 ± 0.1	0.10 ± 0.06	1.56 ± 0.06
**11**	3.2	4.4	0.9	64	1.7 ± 0.2	1.6 ± 0.1	0.27 ± 0.04	1.76 ± 0.05
**12**	4.0	4.6	0.4	84	2.1 ± 0.3	3.6 ± 0.3	1.30 ± 0.07	1.63 ± 0.04
**13**	7.5	11.3	2.0	75	3.4 ± 0.2	3.0 ± 0.1	0.71 ± 0.04	1.64 ± 0.02
**14**	6.1	15.5	4.1	84	2.3 ± 0.4	1.8 ± 0.3	0.62 ± 0.18	1.57 ± 0.09
**15**	8.8	18.2	4.1	79	3.1 ± 0.2	2.0 ± 0.2	0.50 ± 0.06	1.58 ± 0.04
**16**	5.1	8.2	1.5	83	2.4 ± 0.2	2.8 ± 0.2	0.90 ± 0.06	1.63 ± 0.04
**17**	4.9	7.8	1.7	70	2.8 ± 0.3	2.1 ± 0.2	0.12 ± 0.04	1.79 ± 0.06
**18**	5.1	13.7	6.7	67	2.7 ± 0.2	0.83 ± 0.1	−0.06 ± 0.09	1.49 ± 0.05
**19**	6.1	11.3	3.6	72	2.3 ± 0.5	1.3 ± 0.2	0.32 ± 0.09	1.55 ± 0.08
**20**	5.8	10.3	2.3	87	2.5 ± 0.2	2.5 ± 0.2	0.72 ± 0.04	1.64 ± 0.03
**21**	5.3	11.7	3.8	52	2.6 ± 0.2	1.4 ± 0.1	0.11 ± 0.03	1.63 ± 0.04
**22**	5.6	10.2	2.5	65	2.7 ± 0.4	2.1 ± 0.2	0.41 ± 0.09	1.65 ± 0.07

**Table 3 entropy-24-00745-t003:** The studied shock crossings with Alfvénic to linear and non-linear whistler Mach number ratios, maximum electric field strength measured across the shock and electron kinetic entropy change.

#	$M_{A}/M_{\mathrm{wh}}$	$M_{A}/M_{\mathrm{whn}}$	${\left|E\right|}_{\max}\;\left(\mathrm{mV}/m\right)$	$\Delta S_{e}/k_{B}$
**1**	0.74	0.56	739	0.59 ± 0.05
**2**	1.2	1.01	224	−0.06 ± 0.06
**3**	1.9	1.6	178	0.51 ± 0.13
**4**	44	29	121	1.44 ± 0.09
**5**	1.05	0.85	271	0.39 ± 0.04
**6**	1.5	1.2	121	0.31 ± 0.12
**7**	28	34	130	0.10 ± 0.08
**8**	0.44	0.34	716	0.75 ± 0.07
**9**	2.2	2.3	192	1.27 ± 0.09
**10**	0.42	0.33	117	0.10 ± 0.06
**11**	0.47	0.37	143	0.27 ± 0.04
**12**	2.0	2.7	482	1.30 ± 0.07
**13**	2.1	1.8	294	0.71 ± 0.04
**14**	7.2	7.2	82	0.62 ± 0.18
**15**	4.4	4.1	261	0.50 ± 0.06
**16**	3.4	2.6	203	0.90 ± 0.06
**17**	1.06	0.82	293	0.12 ± 0.04
**18**	1.6	1.4	194	−0.06 ± 0.09
**19**	1.7	1.5	441	0.32 ± 0.09
**20**	8.6	8.8	302	0.72 ± 0.04
**21**	0.88	0.72	171	0.11 ± 0.03
**22**	1.12	0.93	231	0.41 ± 0.09

## Data Availability

The MMS data are available at the MMS Science Data Center (https://lasp.colorado.edu/mms/sdc accessed on 1 April 2020).

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
