# Peer review of "Electron Kinetic Entropy across Quasi-Perpendicular Shocks"

_entropy, 2022, doi:10.3390/e24060745_

Round 1
Reviewer 1 Report
Report to the author:
In this entropy-1711797 (titled “Electron kinetic entropy across quasi-perpendicular shocks”) by M. Lindberg, A. Vaivads, S. Raptis, P.-A. Lindqvist, Barbara Giles, Daniel Gershman, the authors investigated the change of electron kinetic entropy across the quasi-perpendicular Earth's bow shock using MMS observations. They also estimate the electron adiabatic index from the observations. This is an interesting work and the electron velocity distribution function is carefully treated. However, we need to clarify some points.
(1) How did the authors determine the peak seen in the electric field spectrogram as the plasma frequency in Section 2.3.4? The authors need to explain since the peak does not necessarily correspond to the plasma frequency.
(2) I understand that the authors focus on electron kinetic entropy. However, readers must be interested in ion kinetic entropy, too. Since the ion kinetic entropy is shown in Fig. 6, it may not be difficult to discuss ion kinetic entropy in the manuscript. I would like to highly encourage the authors to discuss ion kinetic entropy. Or, it is helpful to mention a reason not to need to discuss it in this manuscript.
(3) In panels (a) and (b) of Fig. 6, the colors are not correct. This is because the green color indicates the magnitude of the magnetic field and plasma velocity, but the values become negative.
(4) In panel (g) of Fig. 6, we can see that the electron kinetic entropy decreases in the foot region. I'm curious that this is a common feature among the crossings investigated in this manuscript. If it's so, it is quite valuable to mention it in the manuscript.
Reviewer 2 Report
This manuscript presents a study of electron entropy production and heating through collisionless shock crossings at Earth’s bowshock as measured by the MMS mission. The authors find that entropy production is, within error, positive across all instances and they find an adiabatic index consistent with a monatomic gas with 3 degrees of freedom. Interestingly, the change in entropy is less than previous authors have identified for similar shock crossings, and the adiabatic index is significantly different than some previous studies. For these reasons, the article is compelling and appropriate for publication. However, there are several, mostly minor, issues enumerated below that if addressed would further improve the manuscript. They are presented in the order in which they appear in the manuscript.
1. Since this article has been submitted to a journal with a broad readership, it would be helpful to more clearly define some terms that are peculiar to plasma physics. For example,
a. What is plasma beta, and how are total and electron beta defined?
b. What is a supercritical shock in a plasma?
c. What makes a Whistler Mach number critical?
2. In the discussion concerning the relative values of M_A and the whistler Mach numbers in section 2.2.2, the inequalities, M_A < M_wh and M_A > M_whn, and their physical consequences are reversed. Pedagogically, this is a little confusing, and I would suggest choosing one direction for both inequalities and adjusting their consequences appropriately.
3. Generally, the analysis presented seems very thorough and sound; however, the discussion concerning the correction for secondary electrons and extrapolation to zero energy seems somewhat arbitrary. Regarding the former, as I understand the procedure, the distribution at energies E < 20 eV is adjusted to be a Maxwellian in the solar wind and a flat top in the magnetosheath. To lowest order, that seems reasonable but is impossible to verify. Then, the lowest energies, which are unmeasured, are extrapolated to E = 0 assuming the electrons are isotropic. Also reasonable but unverifiable. These assumptions are particularly worrisome because the resulting distribution produces a density that is 20% too large compared to density derived from measurements of the plasma frequency, suggesting significant error in the secondary removal and extrapolation. The results of these corrections are also compared to data from the MMS data center, and they do not agree.
Have any alternative methods to correct the electron distribution been attempted? Since the MMS data center produces different values, what method do they use? Is it more widely accepted as accurate (or inaccurate)?
4. At the end of section 3, mention is made of comparing the adiabatic index results with Schwartz, which is eventually done in section 4. It would be helpful to state that the comparison is made in the following section to guide the reader.
5. In section 4, it is noted that Parks et al find a somewhat larger value for Delta S, and it is noted that the inconsistency may be due to using a different definition of the entropy. It would be helpful to at least state their definition and why you have chosen a different definition. It would be even better to use their definition for an interval as a point of comparison, e.g., interval 5.
6. In section 4, it is noted that an additional source of error is for intervals that lack measurements of the plasma frequency to cross check the density, followed with a statement that “we expect these error to be small…” In other examples in the manuscript, the error is at the 20% level, which is not small.
7. Figure 7 and the discussion in section 4 suggest that Delta S_e ~ 1 / beta_e^power, and it is noted that it is not obvious such a dependence should hold based on single-fluid, MHD arguments from Rankine-Hugoniot relations given in the appendix. First, 1 / beta_e ~ 1 / T_e, so an inverse power-law relation between Delta S and the upstream temperature should not be surprising based on the relation given in Eq 12. Second, it is not difficult to construct Rankine-Hugoniot relations for a two fluid Plasma, which will produce a relation similar to that plotted in figure A2, but there will be independent ion and electrons relations rather than a single MHD fluid.
8. In section 4, it is noted that Schwartz et al find rather different (factor of two) adiabatic indices for supercritical shocks than the current manuscript, but Pudovkin et al find similar values. It would be helpful to state any notable differences in how Schwartz came to their result versus the method used in the current manuscript, e.g., is it simply lower quality data or did they use a completely different method. Also, are there any more recent examples of computing gamma at shock crossings, especially from MMS data? The cited papers are from 1988 and 1997.
9. The analysis in the manuscript focuses on data from only MMS1. Have the other three spacecraft datasets for these intervals been examined? Are the results consistent across spacecraft?
10. Minor issues with tables and figures:
a) In the caption of table 1 it is stated: “The shock normals are obtained as averages using the three mixed methods described in Paschman and Daly…” This statement suggests the three normals are computed from the mixed methods. However, in the manuscript, it is stated that the normals are determined using two coplanarity approaches and an average of the three mixed methods. Please clarify which methods are used in both the manuscript and caption.
b) In the caption of figure 7e, please specify upstream beta_e for clarity.
11. A few typos I noticed
a) Line 41: it’s —> its
b) Line 148: require —> requires
c) Line 163: spacecraft, it gives —> spacecraft, gives
d) Line 186: the both —> the
e) Line 191: by electron —> by the electron
f) Line 205: 439 km/h —> 439 km/s
g) Line 263: Additional possible —> An additional possible
h) Lines 284 and 285: The right bracket is missing from the expressions for gamma.
i) Lines 319-320: Specify upstream beta_e on both lines for clarity
Round 2
Reviewer 1 Report
I would like to thank the author's sincere response, and it is quite sure now that the manuscript is ready to be published.
Reviewer 2 Report
The authors have well addressed my original concerns, and I now find the manuscript fully appropriate for publication.